# OpenReview forum: "Scalable Oversight for Superhuman AI via Recursive Self-Critiquing"
_ICLR.cc/2026/Conference — ICLR 2026 Conference Desk Rejected Submission_

### Official Review · Reviewer_jtxP · 2025-10-27

**Soundness:** 3
**Presentation:** 2
**Contribution:** 3
**Rating:** 6
**Confidence:** 3

**Summary:**

This paper addresses a critical challenge in AI alignment: how to provide effective oversight when AI systems surpass human capabilities. The authors propose and validate two key hypotheses: (1) critique of critique is easier than direct critique, extending the principle that "verification is easier than generation"; and (2) this difficulty relationship holds recursively, allowing higher-order critiques to provide a more tractable supervision pathway. Through comprehensive Human-Human, Human-AI, and AI-AI experiments across multiple tasks, the paper demonstrates that recursive self-critiquing shows promise as a scalable oversight method for superhuman AI systems.

**Strengths:**

The paper demonstrates several notable strengths:

1. Originality: The extension of the "verification is easier than generation" principle to recursive critique represents a creative and novel approach to the scalable oversight problem. The recursive formulation provides a new pathway for supervision when direct evaluation becomes infeasible.

2. Quality of Experiments: The experimental design is comprehensive and methodical. The progression from Human-Human to Human-AI to AI-AI experiments creates a robust evidence chain. The inclusion of multiple tasks with varying cognitive demands strengthens the generalizability of the findings.

**Weaknesses:**

1. Limited Task Diversity: Although the paper includes five different tasks, they are all in the domain of academic-style problems (language comprehension, mathematics, logical reasoning). The effectiveness of recursive critique in more creative or open-ended domains remains unexplored. Including tasks with more subjective evaluation criteria would strengthen the generalizability of the findings.

2. Scalability Concerns: The paper doesn't adequately address potential scalability issues with recursive critique. As the depth of recursion increases, there may be diminishing returns or compounding errors that aren't explored in the current experiments.

3. Model Limitations: The AI-AI experiments focus primarily on models from the same family (Qwen2.5). The effectiveness of recursive critique across more diverse model architectures remains unclear.

**Questions:**

1. How would recursive critique perform in more creative or subjective domains like art generation, creative writing, or philosophical reasoning where there may not be clear "correct" answers?

2. Is there an optimal depth for recursive critique beyond which the benefits diminish? The paper shows improvements up to C3, but what about C4 or higher?

3. How sensitive is the effectiveness of recursive critique to the quality of the initial critiques? If early critiques are flawed, do these errors compound through the recursion?

---

### Official Review · Reviewer_gPuL · 2025-10-31

**Soundness:** 2
**Presentation:** 2
**Contribution:** 2
**Rating:** 4
**Confidence:** 3

**Summary:**

The paper proposes a “recursive self-critiquing” protocol for scalable oversight: given two initial responses to a question, a critic compares them (C1), then a higher-order critic compares the critics (C2), and so on (C3). The authors argue that “critique of critique” should be easier than direct critique and validate this via Human–Human experiments on multiple-choice tasks (CET-6, Gaokao Chinese/Math, KAOGONG), reporting consistent gains in accuracy and confidence and stable or reduced completion time as depth increases. Overall, the paper positions recursive self-critiquing as a promising but not yet universally effective mechanism for supervision when direct evaluation is hard.

**Strengths:**

1) The problem in study is interesting and important. The paper tackles scalable oversight when direct human evaluation becomes infeasible, articulating the hypothesis that “critique of critique” is easier than critique and exploring its implications for alignment and supervision workflows  .
2) The paper is well written and easy to follow. The protocol is clearly specified (R → C1 → C2 → C3) and concise.
3) Both AI and human studies are conducted.

**Weaknesses:**

1) The scope and strength of AI evaluations are limited. Human–AI experiments consider Qwen2.5-7B and 72B, and supplemental AI–AI experiments use Gemma2-9B and Qwen2.5-14B; frontier or specialized reasoning models are not included.
2) Possible test-time scaling confounds remain. Although the paper compares against effort-equivalent majority voting, token-level and call-level budgets can still differ across stages and models. I wonder whether the performance lift purely come from test-time scaling, e.g., as reasoning models like DeepSeek R1, Qwen3, or OpenAI's o series, with the deep thinking budget increased, the performance become better.
3) Incremental novelty relative to multi-pass and self-critique paradigms. The approach echoes established ideas such as self-consistency voting and debate-like comparative judging; the added contribution is the explicit recursive framing and human study design. The related work section acknowledges these connections (e.g., self-consistency; debate), but the paper would benefit from more direct, head-to-head comparisons on shared benchmarks to clarify what is truly new in capability or reliability .
4) Heavy reliance on multiple-choice tasks constrains external validity. The core evidence comes from MCQ datasets, where comparative judgment and voting are particularly effective; it remains unclear how well the method transfers to open-ended, long-form, or tool-augmented tasks where correctness is not easily discretized.

**Questions:**

Please refer to my questions posted in the weaknesses section. Additionally, I wonder that is the fundamental difference between deep thinking in large reasoning models with your higher-order critiquing method.  I know these two methods have totally different surface, but think about this: in the deep-thinking process, the LLM can role play roles of different critiques, which has the potential to bring higher-order critiquing.

---

### Official Review · Reviewer_mj9z · 2025-11-01

**Soundness:** 4
**Presentation:** 3
**Contribution:** 3
**Rating:** 8
**Confidence:** 3

**Summary:**

This paper investigates the concept of recursive critiquing as both a method for improved human oversight of models, and a potential method for improved model self-supervision. It extends the idea of RLHF, where a critic (human or judge model) critiques two candidate outputs from a model (C1), to critique-of-critique (C2), where the critic critiques two candidate critiques of the original two responses, to critique-of-critique-of-critique (C3), and so on. It investigates the effectiveness of these varying levels of recursive critiquing across four datasets, in three different scenarios: human-human, human-AI, and AI-AI. In the human-human experiment, the authors investigate whether humans performing recursive critiques of human responses and critiques, perform better than a comparable number of majority votes. They find that they consistently do so. In the human-AI experiment, they investigate whether humans can recursively critique the outputs (and critiques) of a model whose baseline performance exceeds theirs on the task, again finding that this is effective. Finally, in the AI-AI experiment the paper investigates whether recursive self-critiquing is effective for self supervision. They find that this paradigm is effective in weak-to-strong supervision, less so in strong-to-weak supervision.

**Strengths:**

- **Novel, interesting idea:** The idea of extending response critique to critique-critique and recursively onward is interesting and novel.

- **Consistently positive results**: The results are consistently positive for human-human and human-AI recursive critiquing, and provide some interesting results for AI self-supervision, opening up possibilities for bootstrapping stronger models from weaker ones.

- **Diverse datasets:** The paper covers a diverse range of datasets, including English proficiency, reading comprehension, math, and logical reasoning

**Weaknesses:**

- **Clarity of tables 1 and 2:** If I understand correctly, the C^2 and C^3 result in the majority voting column in each of these tables is just copied from the accuracy column in order to allow a vertical comparison. Rather than doing this, you should just leave that part of that column blank. If I am misunderstanding, then I don’t know what C^2 or C^3 is for the Majority Voting column or why it is identical  to the accuracy column.

- **Small(ish) models**: A very minor weakness: the human-AI experiment uses Qwen-7B for 3 datasets and Qwen-72B for the fourth. These are relatively small and uncapable compared to frontier models like current versions of DeepSeek or GPT. It’d be interesting to see if the result holds for the most powerful available models.

**Questions:**

No particular questions.

---

### Official Review · Reviewer_EtMb · 2025-11-06

**Soundness:** 3
**Presentation:** 2
**Contribution:** 2
**Rating:** 6
**Confidence:** 3

**Summary:**

This paper focuses on the scalable oversight (sometimes also regarded as superalignment) of LLMs, and proposes a novel self-critiquing method, called, recursive self-critiquing, based on two fundamental assumptions: i) Critique of critique can be easier than critique itself and ii) This difficulty assumption recursively holds. To verify these assumptions, the authors conducted comprehensive experiments and analysis, including: i) human-human critique on China’s National College Entrance Examination test items; ii) human-AI critique and iii) AI-AI Critique on the DeepScaleR dataset. Through these experiments, the authors verify their assumptions hold for human-human/AI interactions, and further demonstrate Recursive self-critiquing benefits in weak-to-strong supervision.

**Strengths:**

1. This paper focuses on a very important topic, scalable oversight, and proposes an interesting and insightful assumption: Critique of critique can be easier than critique itself

2. The authors conducted extensive experiments to verify these assumptions. Particularly, I appreciate the human and human-AI interactions study.

3. On the DeepScaleR dataset, the improvement is significant and consistent.

**Weaknesses:**

1. The effectiveness of the proposed method is quite limited. In the main body, only the DeepScaleR dataset is used. Moreover, I found in Appendix C, popular datasets for scalable oversight, e.g., GPQA and MMLU, are actually used and compared, but there is no significant/consistent improvement. This questions the generalization performance of the proposed method.

2. Besides majority voting, the authors didn’t compare any other scalable oversight baselines, nor did they justify the reason. For example, the sandwiching method [1], which is similar to the human-AI Critique, and the debate based methods [2], which is similar to the AI-AI Critique.

3. Many designs are quite hasty, lacking justification and explanation:

    a. Why the first critique evaluates pairs of candidate responses, instead of each candidate response or all of them (line 128)?

    b. Why did you use many reading comprehension questions for human-human/AI experiments, which might be too easy to LLMs, instead of other scientific ones?

    c. In Table 3, why did AI conduct critique and C2, and humans only di C3?

4. Some experiments designs are not convincing. For example, what would happen if you keep increasing the chain length of critique beyond 3?

Reference:

[1] Bowman et al., Measuring Progress on Scalable Oversight for Large Language Models. 2022.

[2] Kenton et al., On scalable oversight with weak LLMs judging strong LLMs. 2024.

**Questions:**

N/A

**Details Of Ethics Concerns:**

The is a large-scale human study but it's unclear whether they passed IRB. Besides, some information, e.g., compensation for human annotators, is also missing.

---

### Note · Program_Chairs · 2025-11-17
**Submission Desk Rejected by Program Chairs**

This paper had authors names in a footnote during the review period (a revision has since been posted); it consequently must be desk rejected.